# Electronic Transport Properties in a One-Dimensional Sequence of Laser-Dressed Modified Pöschl-Teller Potentials

**DOI:** 10.3390/nano15131009

**Published:** 2025-06-30

**Authors:** Carlos A. Dagua-Conda, John A. Gil-Corrales, Miguel E. Mora-Ramos, Alvaro L. Morales, Carlos A. Duque

**Affiliations:** 1Grupo de Materia Condensada-UdeA, Instituto de Física, Facultad de Ciencias Exactas y Naturales, Universidad de Antioquia UdeA, Cl 70 No. 52-21, Medellín 050010, Colombia; alvaro.morales@udea.edu.co; 2Instituto Tecnológico Metropolitano (ITM)-Institución Universitaria, Facultad de Ciencias, Campus Fraternidad, Calle 73 No. 76A-354 Vía al Volador, Medellín 050034, Colombia; johngil1275@correo.itm.edu.co; 3Centro de Investigación en Ciencias-IICBA, Universidad Autónoma del Estado de Morelos, Av. Universidad 1001, Cuernavaca 62209, Morelos, Mexico; memora@uaem.mx

**Keywords:** electronic transport properties, Pöschl-Teller potential, current density–bias voltage characteristics, structural parameters, non-resonant intense laser field

## Abstract

Modifying the potential profiles in low-dimensional semiconductor heterostructures changes the confinement of particles, impacting the electronic transport properties. In this work, we study the electronic transport properties of a modified Pöschl-Teller double-barrier potential heterostructure of GaAs/AlGaAs, and for a similar double-barrier system including a Pöschl-Teller well between the barriers. For these two configurations, we calculated the current density–bias voltage characteristics, varying barrier and well half-width, the separation between barriers, and the depth of the central well. Additionally, the application of a non-resonant intense laser field. Our results show a redshift in the electronic transmission with increasing barrier separation, and a decrease in the area under the electronic transmission curve with the increase in the half-width of the barriers for both models. The characteristic current density-bias voltage curves in both models exhibit negative differential resistance, with tunable peaks that can be varied through changes in structural parameters and the external laser field.

## 1. Introduction

The quantum confinement effect in low-dimensional systems such as quantum dots (QDs), quantum wells (QWs), quantum wires, quantum well tubes, and superlattice structures have provided during the last decade one of the most attractive platforms for studying the electronic transport properties and involve applications such as high-performance and low-power transistors, quantum information, sensing and metrology, and optoelectronic devices [1,2,3,4,5]. Typically, quantum confinement in such systems can be engineered by two main effects: one is the structural parameters, where the shape and size of the system are changed [6,7]. The other is through external fields, which alter the electronic structure by changing the distributions of energy levels and wave functions of the confined particles [8,9].

One-dimensional confinement in a double-barrier quantum-well (DBQW) structure produces quasi-bound states in the well region [10], which provide the basis for the resonant tunneling (RT) process. It is well-known that the RT in a DBQW structure occurs when the energy of an incident electron aligns with a quasi-bound state within the well. This alignment significantly increases the probability of the electron tunneling through the entire structure [11,12]. Under external bias, the lowest occupied energy levels in the emitter region drop below the quasi-bounded states in the heterostructure, resulting in negative differential resistance (NDR), which is observed when a rise in bias voltage produces a proportional decrease in current (i.e., dI/dV<0). This effect is fundamental in modern technological applications, such as communication and sensing. Interestingly, this fact motivates us to think whether it is possible to control the electronic transport properties by carefully designing a semiconductor structure with novel functionalities.

DBQW AlGaAs/GaAs resonant tunneling diodes (RTDs) have been intensively investigated due to their tunable NDR behavior [13] and compatibility with III–V epitaxial growth techniques [14]. These experimental techniques enable precise control of the growth of individual AlGaAs barriers separated by a GaAs quantum well. In AlGaAs/GaAs RTDs are observed peak-to-valley current ratios (PVCRs) ranging from 3:1 to over 10:1, with peak current densities of $10^{2}$–$10^{4}$ A/cm${\,}^{2}$ depending on barrier thickness and doping concentrations [15,16,17]. Other studies explored relevant RTD applications. For example, in an experimental study, Muttlak et al. [18] designed a DBQW RTD made of InGaAs/AlAs for THz applications using five different device structures. Their results demonstrate a high peak-to-valley current ratio for a device with a very low current density, and show that reducing barrier and well thicknesses significantly increases the current density at the cost of increasing peak voltages. Romeira et al. [19] show a RTD photo-detector (RTD-PD) device that depends on the bias voltage to operate in non-oscillating or oscillating regimes. Their results demonstrate efficient detection of gigahertz (GHz) modulated optical carriers and optical control of a RTD GHz oscillator. Other studies have reported applications such as mid-infrared sensor [20] and high-speed communication [21].

Numerical simulations of DBQW RTDs typically assume rectangular potential profiles for barriers and wells; however, these profiles introduce non-physical artifacts into the simulations. Recently, potential profiles are simulated using a one-dimensional Pöschl-Teller potential that involves a hyperbolic function, which replaces the stepwise function with smoothly varying interfaces [22,23,24]. The Pöschl–Teller potential has analytic solutions for bound-state energies and transmission coefficients, eliminating the non-physical results introduced by rectangular approximations. In a theoretical study, Batı et al. [25] determined the influence of the structural parameters and a non-resonant intense laser field on the resonant tunneling properties in hyperbolic Pöschl-Teller double quantum barrier structure. Their results show that the structural parameter and the laser field allow for control over the electronic spectrum, inducing notable red or blue shifts in the first resonant peak. Furthermore, Rodríguez et al. [26] study the electronic properties of a one-dimensional disordered finite chain of Pöschl-Teller potentials. Their results reveal a fractal distribution of states within specific energy levels, and two types of resonances emerge for the uncorrelated case.

In this work, we study the electronic transport properties in the ballistic regime (without electron scattering) within the effective mass approximation, through a one-dimensional sequence of Pöschl-Teller potentials, considering the influence of variations in the structural parameters and the application of a non-resonant intense laser (nIL) field. Based on a DBQW AlGaAs/GaAs RTD, two structural configurations, denoted as model 1 and model 2, are proposed. Model 1 consists of two Pöschl-Teller barriers separated by a distance $L_{w}$. Model 2 consists of two Pöschl-Teller barriers and an additional Pöschl-Teller well located symmetrically in the center of the barriers [Figure 1]. Our results demonstrate spectral shifts in the electronic transmission, both red-shift and blue-shift, in both models when modifying the structural parameters. Also, we observed an additional resonance state due to the presence of the central Pöschl-Teller well. Under the effect of a nIL field, the potential profiles in both models are dressed, resulting in modifications of the effective height of the barriers and the depth of the wells. These changes in potential profiles induce significant variations in the probability of electronic transmission. By modifying the current density-bias voltage characteristic, which exhibits tunable NDR behavior at low voltages. The calculated peak-to-valley current ratios (PVCRs) are relatively low. However, future work could employ these models and incorporate modulated doping techniques to optimize the PVCR values, thus increasing their potential for applications in electronic devices.

This paper is organized as follows. Section 2 shows the schematic representation of the sequence of modified Pöschl-Teller potentials for the two proposed models. Subsequently, the Schrödinger equations under the effect of a nIL field. Finally, expressions related to the electronic tunneling current density are provided. Section 3 presents and discusses our main results. Section 4 shows our main conclusions and remarks.

## 2. Theoretical Framework

### 2.1. The Modified Pöschl-Teller Potential

The one-dimensional time-independent Schrödinger equation in effective mass aproximation is described in the growth direction of the heterostructure (*x*-coordinate) as follows:(1)$$\left[-\frac{\hbar^{2}}{2m^{*}}\frac{d^{2}}{dx^{2}}+V_{j}\left(x\right)\right]\psi_{n}\left(x\right)=E_{n}\psi_{n}\left(x\right)\;,$$
where $m^{*}$ is the effective mass and $E_{n}$ are the corresponding electron eigenvalues of the *n*-th eigenfunction $\psi_{n}$. According to [27], the confinement potential $V_{j}$ is determined from the assumption of a hyperbolic secant function for the ground state wave function $\psi_{0}\left(x\right)=sech(x)$. The functional form of the confinement potential is described by:(2)$$V_{j}\left(x\right)=-\frac{\hbar^{2}}{m^{*}}\frac{\Lambda_{k}}{\cosh^{2}\left(\frac{x}{\sigma_{k}}\right)}\;.$$
The parameter $\Lambda_{k}$ determines the height or depth of the potential profile and $\sigma_{k}$ the half-width of the barriers and wells.

We consider two structural configurations for the one-dimensional sequence of Pöschl-Teller potential profiles, which we will call model 1 and model 2, shown in Figure 1, given by the following equations, respectively: (3)$$V_{1}\left(x\right)=\left\{\begin{array}{cc} -eFx+\frac{\Lambda_{b}}{\cosh^{2}\left(\frac{x+(L_{w}+5)/2}{\sigma_{b}}\right)}, & L_{1}\leq x\leq L_{2}\;,\\ -eFx, & L_{2}<x<L_{3}\;,\\ -eFx+\frac{\Lambda_{b}}{\cosh^{2}\left(\frac{x-(L_{w}+5)/2}{\sigma_{b}}\right)}, & L_{3}\leq x\leq L_{4}\;, \end{array}\right.$$
and(4)$$V_{2}\left(x\right)=\left\{\begin{array}{cc} -eFx+\frac{\Lambda_{b}}{\cosh^{2}\left(\frac{x+(L_{w}+5)/2}{\sigma_{b}}\right)}, & L_{1}\leq x\leq L_{2}\;,\\ -eFx-\frac{\Lambda_{w}}{\cosh^{2}\left(\frac{x}{\sigma_{w}}\right)}, & L_{2}<x<L_{3}\;,\\ -eFx+\frac{\Lambda_{b}}{\cosh^{2}\left(\frac{x-(L_{w}+5)/2}{\sigma_{b}}\right)}, & L_{3}\leq x\leq L_{4}\;, \end{array}\right.$$
where *e* is the electron charge, the bias voltage $V_{\mathrm{bias}}\left(x\right)=eFx$ correspond to a uniform electric field *F* (V/m) that generate a linear potential. The half-width of the barrier (well) is denoted $\sigma_{b}$ ($\sigma_{w}$), and the height (depth) of the barrier (well) is denoted $\Lambda_{b}$ ($\Lambda_{w}$). In model 1 (Figure 1a), the confinement potential [Equation (Equation 3)] is composed for two Pöschl-Teller barriers located in the intervals $\left[L_{1},L_{2}\right]$ and $\left[L_{3},L_{4}\right]$, separated by a distance $L_{w}$. In model 2 (Figure 1b), the confinement potential [Equation (Equation 4)] is similar to model 1; however, a Pöschl-Teller well is introduced in the interval $\left[L_{2},L_{3}\right]$ (which coincides with the separation between the barriers $L_{w}$).

### 2.2. The Laser-Dressed Modified Pöschl-Teller Potential

The one-dimensional, time-independent Schrödinger equation, within the effective mass approximation and under the influence of an external electric field and a non-resonant intense laser (nIL) field polarized along the *x*-axis, takes the following general form: (5)$$\left[-\frac{\hbar^{2}}{2m^{*}}\frac{d^{2}}{dx^{2}}+\langle V_{j}\left(x,\alpha_{0}\right)\rangle \right]\Psi_{n}\left(x\right)=E_{n}\Psi_{n}\left(x\right)\;.$$
The effect of the nIL field is included in the so-called laser-dressed potential $V_{j}\left(x,\alpha_{0}\right)$ using the Kramers-Henneberger transformation [28], the laser-dressing parameter $\alpha_{0}\equiv eA_{0}/\left(m^{*}\omega \right)$ involves the amplitude $A_{0}$ and the non-resonant frequency of the electric field propagated in the $\hat{x}$ direction [29,30].

Based on Floquet theory, we have assumed that the solution to Equation (Equation 5) under a monochromatic field is given by [31,32]: $\Psi_{n}\left(x\right)=e^{-E_{n}t/\hbar }\sum_{\kappa }\psi_{\kappa }^{n}\left(x\right)e^{-i\kappa \omega t}$, with $E_{n}$ the Floquet quasi-energy. Therefore, by expanding in Fourier’s series and using the Chebyshev polynomials of the first kind $T_{\nu }\left(x\right)$, at high frequencies, the laser-dressed potential can be written as the average of the oscillating potential function, given by [30,33]: (6)$$\langle V_{j}(x,\alpha_{0})\rangle =\frac{\omega }{2\pi }\int_{0}^{2\pi /\omega }V_{j}\left(x+\alpha_{0}\sin\left(\omega t\right)\right)dt\;.$$

### 2.3. The Electronic Tunneling Current Density

We numerically solved the differential Equation (Equation 5) using the finite element method (FEM), including open boundary conditions, implemented in COMSOL Multiphysics version 6.1 [34,35]. The solution corresponds to *x*-dependent probability amplitudes.

In theory, in any region of the structure, the complete wave function is the superposition of the incoming and reflected waves with associated probability amplitudes at each point on the *x*-axis. The electron tunneling through the system can be calculated using the electron transmission function T(E), which is the modulus of the ratio between the probability amplitudes of the transmitted and incident waves, respectively.

The current density-bias voltage characteristics of the system can be calculated by using the Landauer formula [36], which for a macroscopically large system in the transverse direction takes the form,(7)$$J\left(V_{\mathrm{bias}}\right)=\frac{em^{*}}{2\beta \pi^{2}\hbar^{3}}\int_{0}^{\infty }T\left(E,V_{\mathrm{bias}}\right)\ln\left(\frac{1+e^{\beta (E_{F}-E)}}{1+e^{\beta (E_{F}-E-V_{\mathrm{bias}})}}\right)dE\;,$$
where $E_{F}$ is the Fermi level, $V_{\mathrm{bias}}$ the bias voltage applied between the emitter and collector terminals of the system, $\beta =1/k_{B}T$, $k_{B}$ is the Boltzmann constant, T=300 K is the temperature and $T(E,V_{\mathrm{bias}})$ is the electronic transmission with an explicit dependence on the energy of the incident electron and the external bias voltage.

## 3. Results and Discussion

We consider a one-dimensional sequence of Pöschl-Teller potentials. The system has barrier (Al${\,}_{x}$Ga${\,}_{1-x}$As) and well (GaAs) regions with Al concentration of x=0.3. The effective mass of GaAs is $m^{*}=0.067m_{0}$, where $m_{0}$ is the electron mass, this value has been measured experimentally in various studies [37,38] and also calculated theoretically, primarily using $\vec{k}\cdot \vec{p}$ theory [39,40]. The empirical band offset ratio at GaAs/Al${\,}_{x}$Ga${\,}_{1-x}$As interfaces is 6:4, which assigns a 60% conduction band offset and 40% valence band offset [41]. Thus, the height of the barrier is $\Lambda_{b}=\Xi \left(E_{g}^{AlGaAs}-E_{g}^{GaAs}\right)=228$ meV with Ξ=0.60. The structural parameters such as barriers separation ($L_{w}$), barrier half-width ($\sigma_{b}$), well ($\sigma_{w}$), and well depth ($\Lambda_{w}$) vary. Furthermore, an external electric field and a nIL field are applied to the system.

Figure 2 shows the two potential profiles studied, firstly, in Figure 2a we see the model 1 [solid line Figure 1] characterized by a height $\Lambda_{b}$, a mean width $\sigma_{b}$ and separation between barriers $L_{w}$ (see black curve); on the other hand, in Figure 2b we see the model 2 [dashed line Figure 1] characterized by a depth $\Lambda_{w}$, a mean width $\sigma_{w}$. These two models have included the effect of a nIL field characterized by the laser-dressing parameter $\alpha_{0}$. The black curves correspond to potentials without the laser effect ($\alpha_{0}=0$), and the other curves show the modifications that the potential undergoes as the laser parameter increases to a maximum value of $\alpha_{0}=4.0$ nm. Note how as this parameter increases, the height and shape of the barriers decrease significantly, as does the Pöschl-Teller well. Later in the work, we study how these modifications affect the electronic transmission and transport properties.

Figure 3 shows the transmission probability as a function of the energy of the incident electrons for three different $L_{w}$, where $L_{w}$ corresponds to the separation between the Pöschl-Teller barriers, Figure 3a is for model 1 and Figure 3b is for the model 2 [see Figure 1]. In Figure 3a we see how as $L_{w}$ grows, the first quasi-steady state presents a redshift, which corresponds to can be explained by the increase in the central well width which decreases the confinement and causes the eventual inclusion of new states in this region, generating a decrease in the energy of the lowest state. Also, note that with an increase in $L_{w}$, the half-width of that state becomes smaller; this is due to the decrease in the imaginary part of the energy corresponding to said state because it is at a lower energy value (by the redshift), the quasi-steady state resembles what would be a steady state in the central well. On the other hand, in Figure 3b we see that with the inclusion of the additional central Pöschl-Teller well, the lowest state for $L_{w}=2.5$ nm is located very close to 0 eV, that is, close to the bottom of the band in the emitter, for $L_{w}=5.0$ nm and $L_{w}=7.5$ nm this state no longer appears since it is located inside the Pöschl-Teller well, in this sense, by including a central Pöschl-Teller well it is possible to suppress the contribution of the first quasi-stationary state to the transport properties. Only higher states are necessary for this configuration; further explanation will come later.

Figure 4 shows the transmission probability as a function of the energy of the incident electrons for three different barriers mean width parameter $\sigma_{b}$, Figure 4a for model 1 and Figure 4b is for the model 2. We can observe that in model 1 [Figure 4a], the resonance energies are blueshifted as the width of the barriers increases. The reason is that increasing the width of the barriers increases the confinement in the central well, leading to a shift of the energies towards higher values. In model 2 [Figure 4b], by including a central Pöschl-Teller well, the first resonant state is shifted in the well region and contains negative energy values, which do not contribute to the transmission coefficient. For the other resonances, the increase in barrier width has a similar effect to that shown in model 1.

Figure 5 shows the effect of the nIL field characterized by the laser-dressing parameter $\alpha_{0}$. Modifications in the potential profiles of the two proposed models are shown in Figure 2, where, as $\alpha_{0}$ increases, in model 1, a decrease in barrier heights is evidenced, followed by the formation of a barrier-well-barrier structure in the regions where the individual barriers are located; this configuration of the potential profiles favors the appearance of new resonant states. Finally, for model 2, the dressing effect is similar to model 1, but the presence of the central Pöschl-Teller well as $\alpha_{0}$ increases its depth, and the states that were initially confined there tend to leave this confinement region. Due to the above, in Figure 5a, we can notice that in model 1 for laser-dressing parameter $\alpha_{0}=4$ nm, there are more resonant states close to each other that can contribute to the transmission coefficient and lead to the appearance of a plateau. In Figure 5b, model 2 presents two effects of great interest as the laser-dressing parameter increases. First, the resonant energies are red-shifted, indicating a decrease in confinement. Second, the appearance of a resonant state at small energies, similar to model 1, suggests that through the laser, we can modify the confinement to achieve a confinement identical to that presented in model 1.

Figure 6 shows the transmission coefficient for different bias voltages. For model 1 (Figure 6a), increasing the bias alters the electron confinement, producing a redshift of the resonance energies. This shift is due to the modification of the confinement, which alters the allowed energy levels for the quantum states. Additionally, we observe an increase in the area under the curve related to the imaginary part of the quasi-stationary states, which describes the time decay rate of the standing wave associated with the state. We can find a higher time decay as the bias increases. As the system polarizes, the resonance energies can reach high enough values to escape the wells and enter the continuum of states. As states enter the continuum of states, the density of states increases significantly, which means that more states are available for the electrons to interact and contribute to a higher scattering rate and thus a larger width of the resonances. Figure 6b shows that model 2 exhibits the same bias-induced redshift observed in Figure 6a. However, it is interesting to note that the bias enables resonance energies close to the Fermi level, which indicates that the confinement is similar to model 1 for large bias values.

Figure 7 shows the current density versus bias voltage (*J*–*V*) characteristics curves for model 1 (a) and model 2 (b), each Figure presents three curves corresponding to three different barriers separations, $L_{w}=$ 2.5 nm (black curve), $L_{w}=$ 5 nm (red curve), and $L_{w}=7.5$ nm (blue curve). The remaining parameters have been set to $\sigma_{b}=0.6$ nm, $\sigma_{w}=0.6$ nm and $\alpha_{0}=0$. Note that for the minimum separation, two peaks (NDR peaks) are evident in the *J*–*V* curves for model 1 (see the black curve in Figure 7a), the first occurs at approximately 0.1 V, and the second at around 0.4 V. The first peak corresponds to the resonant tunneling of the electrons through the first quasi-stationary state (see in Figure 3a the first peak in the black curve), which is located approximately at 50 meV. On the other hand, at high voltages, the second peak in the characteristic curve arises from resonant tunneling through a second state of the continuum. The second peak is significantly larger than the first because it originates from a higher state with a shorter electron lifetime, resulting in an increased current.

Note how as the separation between the wells increases, there is a shift of the current density peaks towards lower voltages, (compare black curve, then red curve and finally blue curve in Figure 7a this arises from the displacement of the quasi-stationary states approaching the bottom of the conduction band and the decrease in magnitude is because as the voltage increases, the system becomes more asymmetric, which decreases the magnitude of the resonant tunneling since the states no longer reach a probability of 100%.

On the other hand, in Figure 7b corresponding to the model 2, for $L_{w}=$ 2.5 nm (black curve) a peak close to 0.45 V but the first peak that was evident in Figure 7a near 0.1 V disappears, this is because said state is located very close to the bottom of the central well in model 1 and by adding a central Pöschl–Teller well between the two barriers (model 2), the state becomes confined within that well and ceases to be a quasi-stationary state to become a stationary state that does not contribute to resonant tunneling and therefore, does not contribute significantly to the current density. As the bias voltage increases, it is evident as in Figure 7a a shift of the peaks towards lower voltages and the appearance of new peaks of NDR, these behaviors are again explained by the shift of the states towards the bottom of the conduction band with increasing bias voltage, followed by a decrease in the probability density of each one and the inclusion of new quasi-stationary states that contribute significantly to the current density.

Figure 8 shows the *J*–*V* characteristic curves for the two models studied, the double-barrier model 1 (Figure 8a) and the double-barrier model 2 including a central well (Figure 8b). In each of the figures, the parameter $\sigma_{b}$ corresponding to the mean width of the barriers has been varied from 0.6 nm (black line and dots), 0.75 nm (red line and dots) and 0.9 nm (blue line and dots). For both models the appearance of NDR is again evident; note that the appearance of the first two current peaks occur for voltages of around 0.05 V and 0.2 V respectively in model 1 (Figure 8a), these local current maxima correspond to the first and second resonant tunneling with the first two quasi-stationary levels located around 0.25 meV and 100 meV, as previously evidenced in Figure 4a.

Similarly, in Figure 8b corresponding to model 2, two distinct current peaks emerge at approximately 0.17 V and 0.4 V. Again, these peaks can be explained from the two resonances with the quasi-stationary states presented in Figure 4b for values of approximately 100 meV and 200 meV. The current peaks for higher voltages have a greater magnitude because they correspond to increasingly higher quasi-stationary states, and therefore have a greater probability amplitude and a much shorter lifetime, which causes the electrons to tunnel much faster through these states. For both models, a decrease in the magnitude of the current peaks is evident as the barrier half-width increases; this is due to the reduction in the area under the electronic transmission curve caused by a decrease in the tunneling probability (compare the area under each curve in Figure 4).

Figure 9 shows the *J*–*V* characteristic curves for the two models (model 1 Figure 1a and model 2 Figure 1b) for different values of the laser-dressing parameter $\alpha_{0}$. For both models, the NDR peaks are evident for values of the laser parameter less than or equal to $\alpha_{0}=1$ nm, for example, note the disappearance of the second resonant peak located approximately at 0.15 V when going from $\alpha_{0}=1$ nm to $\alpha_{0}=2$ nm in model 1, this can be understood by the decrease in the height of the barriers when the value of the laser parameter increases, and this behavior occurs for both models. Note that in model 2 there is an evident peak around 0.4 V for $\alpha_{0}\leq 1$; however, for $1<\alpha_{0}$ this peak is no longer present. The behavior described above allows for the possibility of modulating the electronic transport properties through the heterostructure using the application of a nIL field, which corresponds to an external effect.

The reason for the large values of the laser parameter (compare, for example, the green and orange curves in Figure 9a,b) is because as the value of the laser parameter increases, in both systems the height of the barriers decreases, but, additionally, in model 2 the depth of the central well decreases, to the point where the inner steady state becomes a quasi-steady state resembling model 1 (this behavior was previously evidenced in Figure 5b in the orange curve corresponding to $\alpha_{0}=4$ nm).

Figure 10 shows the *J*–*V* characteristic curves for model 2, varying the depth of the well $\Lambda_{w}$. The remaining parameters have been set at $L_{w}=7.5$ nm and $\alpha_{0}=0$. Note how for the value of $\Lambda_{w}=75$ meV (curve and black dots) a current peak is evident around 0.02 V, this peak corresponds to the resonant tunneling through the first quasi-stationary state (for this well width there is no confined state), this state is evident in the electronic transmission with the first sharper peak (black curve in the inset). For the following widths, $\Lambda_{w}=150$ meV and $\Lambda_{w}=225$ meV, the quasi-stationary state becomes a stationary state inside the central well and does not contribute to the electronic transmission as evidenced in the inset (see blue and red curves), therefore, it also does not contribute to the current density. The following resonance in the current that occurs for a voltage of approximately 0.18 V is equivalent to the three values of $\Lambda_{w}$. The inset shows the amplified curve for low bias voltage to identify the current peak result from the first stationary state for $\Lambda_{w}=75$ meV (dots and black line). In that case, this peak does not appear for the remaining lines. Hence, the quasi-stationary state becomes a stationary state in the central Pöschl-Teller well (this state does not appear in the curve in the box since it corresponds to a negative energy state regarding the potential level used).

Table 1 shows the peak-to-valley current ratio (PVCR) from the *J*–*V* characteristic curves calculated using the relation $\mathrm{PVCR}=I_{P}/I_{V}$, where $I_{P}$ is peak current density and $I_{V}$ is the valley current. We display the first two NDR peaks (PVCR${\,}_{1}$ and PVCR${\,}_{2}$) for each figure, as indicated in the first column, for the studied models and varied parameters. The results presented in Table 1 correspond to a characterization of the system, but do not correspond to an optimization of parameters to maximize the PVCR of the system. According to the results, the maximum PVCR values are obtained for model 1 at the first NDR peak with a parameter of $L_{w}=2.5$ nm, reaching a value of PVCR${\,}_{1}=1.6$, similarly, model 1 at $\sigma_{b}=0.9$ nm, yields the same PVCR for the second resonant peak. Note that in Figure 7b for $L_{w}=2.5$ nm (black curve), the PVCR calculation is not applicable (N/A) because no NDR peaks are present. A similar effect occurs when the laser-dressing parameter reaches values of $\alpha_{0}=2.0$ nm and 4.0 nm.

## 4. Conclusions

Electronic transport properties have been studied and discussed for two models based on GaAs/AlGaAs, the first model corresponds to a double barrier system of the Pöschl-Teller potential and the second model corresponds to the double barrier system including a central Pöschl-Teller well which allows the appearance of stationary states. We study the electronic transport characteristics, analyzing the response of the system to structural variations in the half-width of the barriers, the separation between them, and the depth of the central well. Likewise, we studied the response to a nIL field polarizing the system with an external bias voltage.

The results show a redshift in the electronic transmission for both models as the barrier separation increases. For model 2, this redshift is followed by the emergence of a steady state within the central Pöschl-Teller well. As a consequence of this behavior, the characteristic current density-bias voltage curves for both models exhibit evident peaks of negative differential resistance (NDR) that can be modulated for different values of barrier separation. This behavior arises for lower voltages in model 1. The results concerning the increase in the average width of the barriers indicate a much less significant blueshift than that presented in the previously mentioned case. The most notable result now is the decrease in the area under the electronic transmission curve with the increase in said width; this behavior is similar for both models. For this case, the characteristic current density-bias voltage curves also indicate the presence of NDR, and the current peaks show a decrease in magnitude with the increase in the average width of the barriers. This behavior is expected, as reduced tunneling probability leads to NDR occurring at nearly the same voltages for both models.

The application of an nIL field significantly alters the potential profiles and, thus, the transmission probabilities. In both models, increasing the laser-dressing parameter raises the area under the transmission curve without noticeably shifting its position because the laser field reduces the height of the barriers and includes additional continuum states. This behavior implies that the current density presents significant modifications and a less regular behavior for large values of the laser-dressing parameter. For these parameter values, the current due to higher voltages originates from a non-resonant tunneling effect caused by states with energies higher than the height of the barriers. On the other hand, in model 2, the emergence of a current peak for low voltages was found when the depth of the central well is low; when this depth increases, this peak disappears due to the shift of the resonant quasi-stationary state to the interior of the well and becomes a stationary state that does not allow tunneling.

The results mentioned above demonstrate the possibility of modulating the electronic properties in the heterostructure, such as the positions and magnitudes of the NDR, either through structural changes or by applying an intense non-resonant laser field.

## Figures and Tables

**Figure 1 nanomaterials-15-01009-f001:**
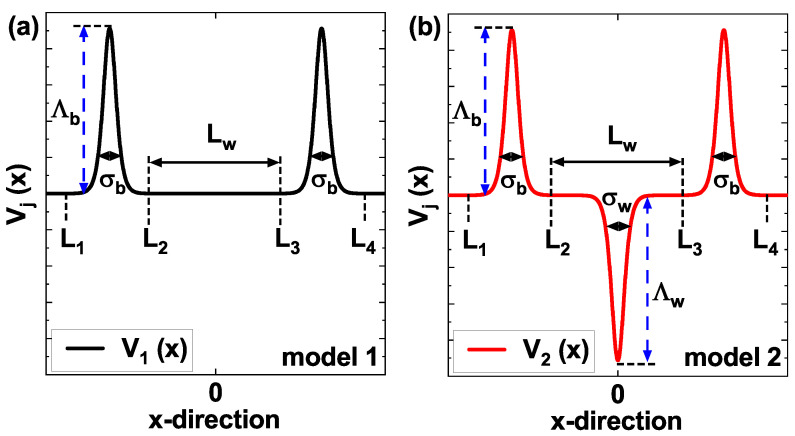
Schematic representation of the one-dimensional sequence of modified Pöschl-Teller potentials without external electric field (F=0). In model 1, two Pöschl-Teller barriers of Al${\,}_{0.3}$Ga${\,}_{0.7}$As separated $L_{w}$ with half-width $\sigma_{b}$ and height $\Lambda_{b}$ (**a**). In model 2, two Pöschl-Teller barriers of Al${\,}_{0.3}$Ga${\,}_{0.7}$As and one central Pöschl-Teller well of GaAs with half-width $\sigma_{w}$ and depth $\Lambda_{w}$ (**b**).

**Figure 2 nanomaterials-15-01009-f002:**
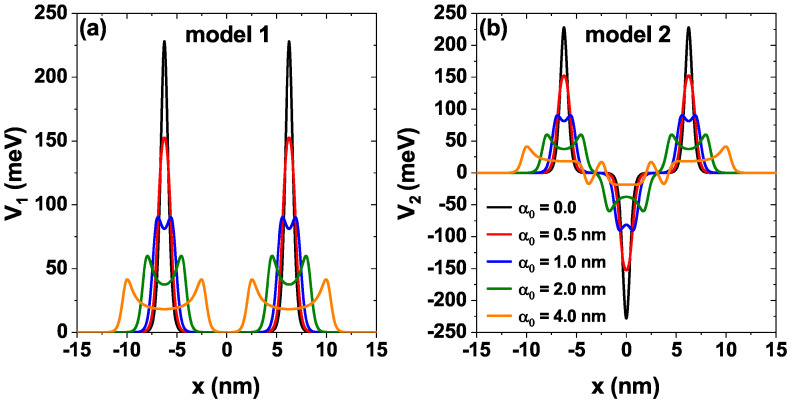
Potential profiles for the sequence of modified Pöschl-Teller of GaAs/Al${\,}_{0.3}$Ga${\,}_{0.7}$As, while varying the laser-dressing parameter $\alpha_{0}=0,\;0.5,\;1.0,\;2.0$, and 4.0 nm of (**a**) model 1 and (**b**) model 2, at $L_{w}=7.5$ nm, $\Lambda_{b}=\Lambda_{w}=228$ meV, $\sigma_{b}=\sigma_{w}=0.6$ nm and F=0.

**Figure 3 nanomaterials-15-01009-f003:**
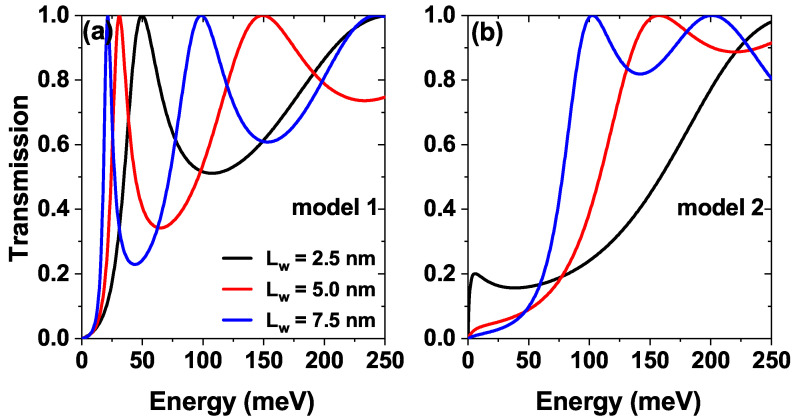
The transmission coefficient as a function of the electron energy for the sequence of modified Pöschl-Teller potentials of GaAs/Al${\,}_{0.3}$Ga${\,}_{0.7}$As, while varying the separation between barriers $L_{w}=2.5,\;5.0$ and 7.5 nm of (**a**) model 1 and (**b**) model 2, at $\sigma_{b}=\sigma_{w}=0.60$ nm, $\Lambda_{b}=\Lambda_{w}=228$ meV and $\alpha_{0}=0$, F=0.

**Figure 4 nanomaterials-15-01009-f004:**
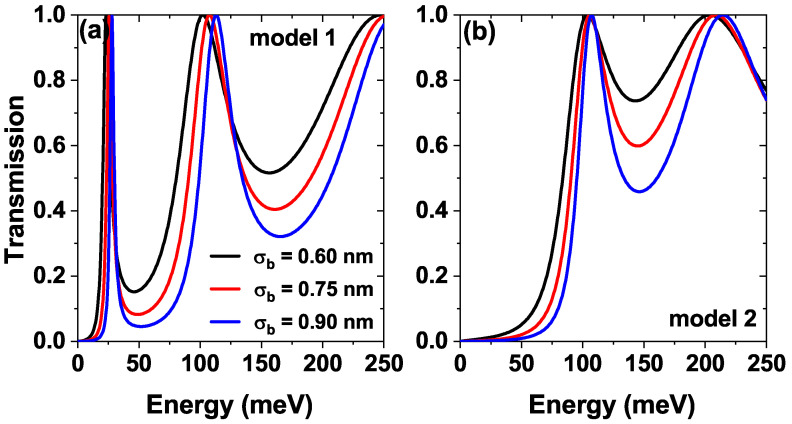
The transmission coefficient as a function of the electron energy, while varying the barrier half-width parameter $\sigma_{b}=0.6,\;0.75$, and 0.90 nm of (**a**) model 1 and (**b**) model 2, at $\sigma_{w}=0.60$ nm, $L_{w}=7.5$ nm, $\sigma_{b}=\sigma_{w}=0.60$ nm, $\Lambda_{b}=\Lambda_{w}=228$ meV and $\alpha_{0}=0$, F=0.

**Figure 5 nanomaterials-15-01009-f005:**
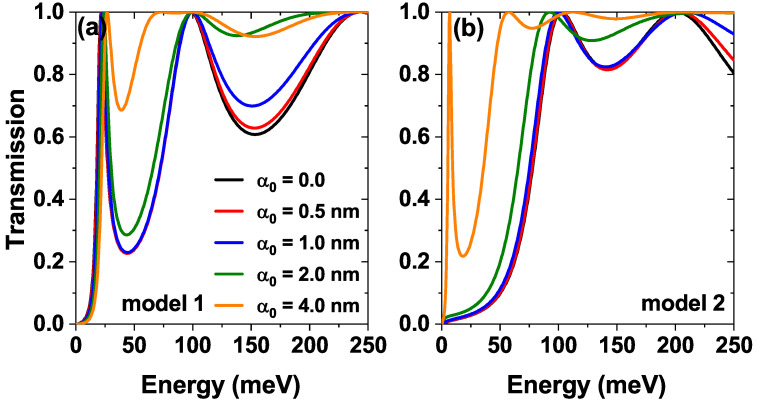
The transmission coefficient as a function of the electron energy, while varying the laser-dressing parameter $\alpha_{0}=0,\;0.5,\;1.0,\;2.0$ and 4.0 nm of (**a**) model 1 and (**b**) model 2, at $L_{w}=7.5$ nm, $\sigma_{b}=\sigma_{w}=0.60$ nm, $\Lambda_{b}=\Lambda_{w}=228$ meV and F=0.

**Figure 6 nanomaterials-15-01009-f006:**
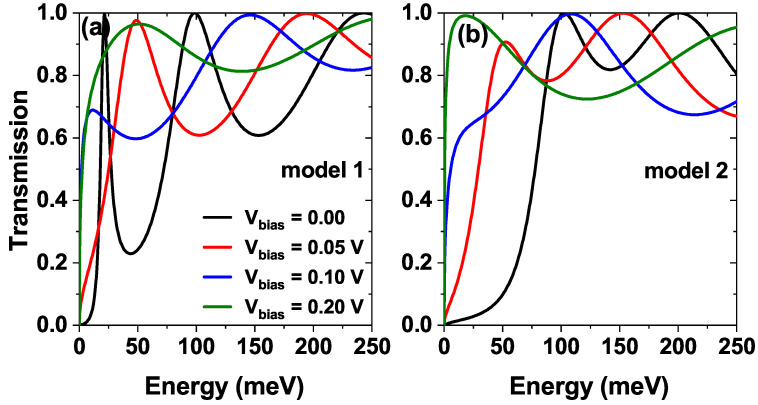
The transmission coefficient as a function of the electron energy, while varying the bias voltage $V_{\mathrm{bias}}=0,\;0.05,\;0.10$ and 0.20 V of (**a**) model 1 and (**b**) model 2, at $L_{w}=7.5$ nm, $\sigma_{b}=\sigma_{w}=0.60$ nm, $V_{b}=V_{w}=228$ meV and $\alpha_{0}=0$.

**Figure 7 nanomaterials-15-01009-f007:**
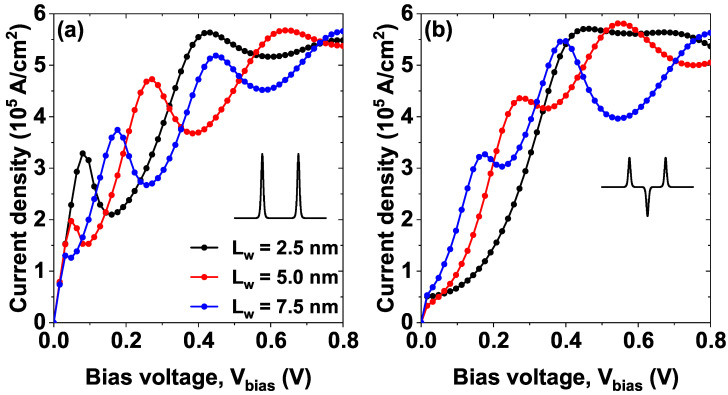
The current density versus bias voltage (*J*–*V*) characteristics curves for the sequence of modified Pöschl-Teller potentials of GaAs/Al${\,}_{0.3}$Ga${\,}_{0.7}$As, while varying the separation between barriers $L_{w}=2.5,\;5.0$, and 7.5 nm of (**a**) model 1 and (**b**) model 2, at $\sigma_{b}=\sigma_{w}=0.60$ nm, $\Lambda_{b}=\Lambda_{w}=228$ meV and $\alpha_{0}=0$.

**Figure 8 nanomaterials-15-01009-f008:**
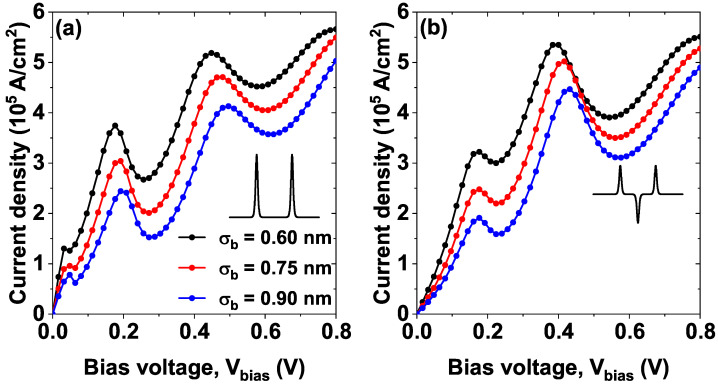
The current density versus bias voltage (*J*–*V*) characteristics curves, while varying the barrier half-width parameter $\sigma_{b}=0.60,\;0.75$, and 0.90 nm of (**a**) model 1 and (**b**) model 2, at $\sigma_{w}=0.60$ nm, $L_{w}=7.5$ nm, $\Lambda_{b}=\Lambda_{w}=228$ meV and $\alpha_{0}=0$.

**Figure 9 nanomaterials-15-01009-f009:**
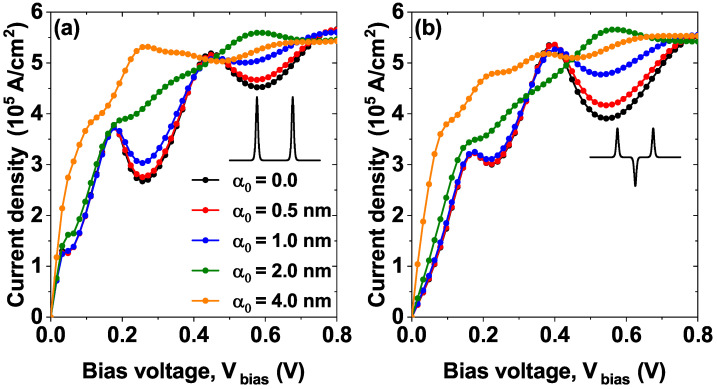
The current density versus bias voltage (*J*–*V*) characteristics curves, while varying the laser-dressing parameter $\alpha_{0}=0,\;0.5,\;1.0,\;2.0$, and 4.0 nm of (**a**) model 1 and (**b**) model 2, at $L_{w}=7.5$ nm, $\sigma_{b}=\sigma_{w}=0.60$ nm and $\Lambda_{b}=\Lambda_{w}=228$ meV.

**Figure 10 nanomaterials-15-01009-f010:**
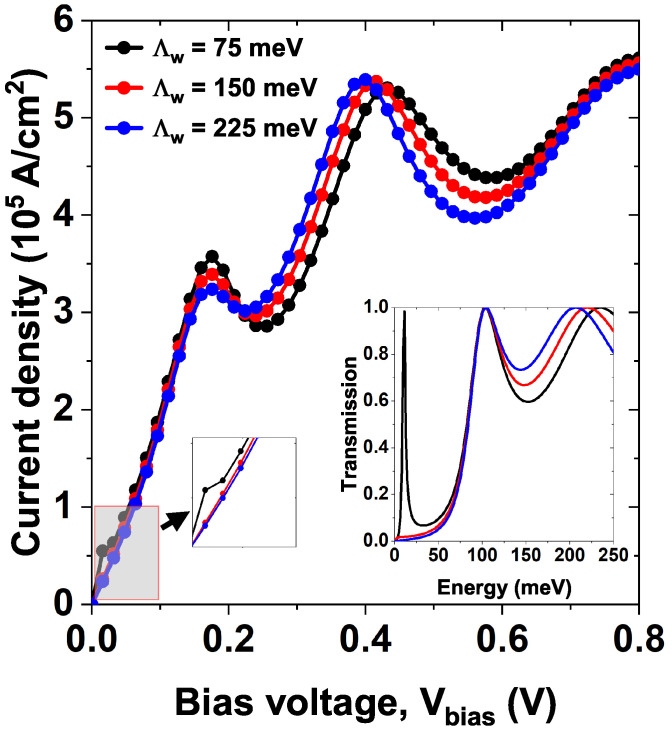
The current density versus bias voltage (*J*–*V*) characteristics curves, while varying the Pöschl-Teller well depth $\Lambda_{w}=75,\;150$, and 225 meV of model 2, at $L_{w}=7.5$ nm, $\sigma_{b}=\sigma_{w}=0.60$ nm, $\Lambda_{b}=228$ meV and $\alpha_{0}=0$. The left and right insets show the curve amplifications and the transmission coefficient.

**Table 1 nanomaterials-15-01009-t001:** Calculated peak-to-valley current ratio (PVCR) for the negative differential resistance peaks in each current-density vs. bias-voltage figures. N/A refers to not applicable.

Figure	Model	Parameter [L, $\alpha_{0}$, σ (nm)] [Λ (meV)]	PVCR${\,}_{1}$	PVCR${\,}_{2}$
		$L_{w}=2.5$	1.6	1.1
Figure 7a	1	$L_{w}=5.0$	1.3	1.3
		$L_{w}=7.5$	1.0	1.4
		$L_{w}=2.5$	N/A	N/A
Figure 7b	2	$L_{w}=5.0$	1.0	1.2
		$L_{w}=7.5$	1.1	1.4
		$\alpha_{0}=0$	1.4	1.1
		$\alpha_{0}=0.5$	1.4	1.1
Figure 8a	1	$\alpha_{0}=1.0$	1.2	1.0
		$\alpha_{0}=2.0$	N/A	N/A
		$\alpha_{0}=4.0$	N/A	N/A
		$\alpha_{0}=0$	1.1	1.4
		$\alpha_{0}=0.5$	1.1	1.3
Figure 8b	2	$\alpha_{0}=1.0$	1.0	1.1
		$\alpha_{0}=2.0$	N/A	N/A
		$\alpha_{0}=4.0$	N/A	N/A
		$\sigma_{b}=0.6$	1.0	1.4
Figure 9a	1	$\sigma_{b}=0.75$	1.0	1.5
		$\sigma_{b}=0.9$	1.3	1.6
		$\sigma_{b}=0.6$	1.1	1.4
Figure 9b	2	$\sigma_{b}=0.75$	1.1	1.4
		$\sigma_{b}=0.9$	1.2	1.4
		$\Lambda_{w}=75$	1.2	1.2
Figure 10	2	$\Lambda_{w}=150$	1.1	1.3
		$\Lambda_{w}=225$	1.1	1.4

## Data Availability

No new data were created or analyzed in this study. Data sharing is not applicable to this article.

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
