# Peer review of "Electronic Transport Properties in a One-Dimensional Sequence of Laser-Dressed Modified Pöschl-Teller Potentials"

_nanomaterials, 2025, doi:10.3390/nano15131009_

Round 1
Reviewer 1 Report
Comments and Suggestions for Authors
In this article, the authors present two models (with and without a central well) using modified Pöschl–Teller potentials to evaluate electronic transport in GaAs/AlGaAs heterostructures. The results clearly demonstrate the tunability of negative differential resistance (NDR) by varying structural and external field parameters. Here are some questions for the authors to consider.
- How realistic are the values chosen for the laser-dressing parameter α0? Could such intensities be achieved without damaging the heterostructure?
- The models in this work assume ballistic transport. How would including scattering (phonons, impurities) or disorder affect the observed NDR behaviors?
- Why was the electron energy range chosen between 0 and 250 meV? Is this range sufficient to capture all relevant resonances?
- Can this approach be extended to 2D or 3D nanostructures with cylindrical or spherical symmetries?
Author Response
Please see the attached file.
Sincerely
Authors

Reviewer 2 Report
Comments and Suggestions for Authors
This manuscript constructs two Pöschl-Teller potential models for AlGaAs/GaAs heterostructure. In Model 1, only AlGaAs is considered, with the potential consisting of two separate barrier potentials. In Model 2, both AlGaAs and GaAs are included, introducing a well between the two barriers. The model potentials are substituted into the Schrödinger equation to solve for the electron transmission function T(E) and the current density-bias voltage (J−V) characteristics. The results analyze the influence of various parameters—including barrier distance (Lw), half-width (σb), laser-dressing parameter (α0), well depth (Λw), and bias voltage (Vbias)—on the transmission function and current density versus bias voltage. These findings provide guidance for tuning the magnitude and position of negative differential resistance (NDR) by adjusting the material structure or applying an intense non-resonant laser field. Here are some questions and suggestions about this paper listed below:
Q1: In Equation (3) and (4), the barrier potential centers are set at (Lw+5)/2 and (Lw-5)/2, can the authors explain why you use 5 here?
Q2: The result at the baseline condition of Lw = 7.5 nm, σb = σw = 0.60 nm, Vb = Vw = 228 meV, α0 = 0, and Vbias = 0 is plotted in all Figure 3, 4, 5, 6. However, the black curve in Figure.4 is different from others at the same condition. In Figure 4, the black curve has lower transmittance (model 1 at 50meV and 150meV, model 2 at 130meV) compared with the others. Could the authors give further explaination?
Q3: In line 120, the laser-dressing parameter formula, α0 ≡ eA0/(m∗ω2) is different from the reference [29], which is given as:
Can the authors explain the reason or provide other references to support your formula?
Q4: In line 147, the authors use the effective mass m* = 0.067m0. How is this effective electron mass calculated, or are there any references to support it?
Q5: The paper briefly mentions using COMSOL for computation and solving. Could the authors provide a more detailed modeling schematic and explain the solution process?

Author Response

(The authors gave the same response as above.)
